# Food Choices of Contemporary Cuisine and Traditional Foods: Effects on Family Ties

**DOI:** 10.3390/nu16234126

**Published:** 2024-11-28

**Authors:** Bushra Yasmeen, Florian Fischer

**Affiliations:** 1School of Sociology, Minhaj University, Civic Center Twp Commercial Area Lahore, Lahore 54770, Pakistan; bushrayasmin.soc@mul.edu.pk; 2Institute of Public Health, Charité—Universitätsmedizin Berlin, 10117 Berlin, Germany

**Keywords:** food choices, contemporary cuisine, structuration theory, Theory of Planned Behavior, kitchen value, family ties

## Abstract

**Background:** Technological advancement has evolved dynamics in the pace of day-to-day life. Economic and social development has introduced new meanings at individual and societal levels. Modernity and development have transformed the social fabric and relationships. Social media has instigated a tremendous multifaceted transformation in lifestyle. An increase in disposable income from prepared food, especially contemporary cuisines, has evolved diversified changes in consumers’ behavior. These changes include trends, perceptions, consumption patterns, and modes of fast food (quality, quantity, tastes), including delivery systems, among all age groups and genders. This study investigated the factors that influence food choices towards contemporary cuisine, the influence of contemporary cuisine food choice on homemade/traditional foods, and how food choices of contemporary cuisines affect family ties. **Methods:** In this exploratory study, a cross-sectional quantitative survey research method was used to obtain the insights of youths. A systematic random sampling procedure was adopted. We recruited at a public sector university of Lahore, namely, the Institute of Social and Cultural Studies, University of the Punjab. The sample size was 260. We used modified versions of the Food Frequency Questionnaire and the Food Choice Questionnaire to assess the food choices of contemporary cuisine, homemade/traditional food, and family ties. The Cronbach’s coefficient alpha level varied from 0.62 to 0.85 among scales. We performed descriptive and inferential statistics (factor analysis and multivariate regression analysis) by using SPSS 23. **Results:** Age and education showed a significant relationship with traditional food. Taste was the only factor identified for food choices of contemporary cuisine. Traditional food and family ties were the factors identified in the analysis. **Conclusions:** Food choices of contemporary cuisines were found to be just for good taste and fun. Homemade food regulates the traditions of traditional food. Cooking and dinning together not only maintained the food choices but also encourage and motivate the connectedness and closeness that strengthen family ties.

## 1. Introduction

The Industrial Revolution introduced tremendous changes. These changes transformed human society globally at all levels. The economic and social development introduced new meanings at individual and societal levels. The transformation introduced new shapes of subjectivity and objective meanings in social life. Technological advancement evolved dynamics in the pace of day-to-day life. Modernity and development have transformed social relationships. Social media instigated a tremendous multifaceted transformation in lifestyle. An increase in disposable income from prepared food, especially contemporary cuisines, evolved diversified changes in consumers’ behavior, like trends, perceptions, consumption patterns, and modes of fast food (quality, quantity, tastes), including delivery systems, among all age groups and genders.

Furthermore, women’s participation and employment in the workforce has changed the inclusion and exclusion criteria of social roles required for the strengthening of social relationships and family ties. Multiple engagements and preoccupations of employed women profoundly affected their roles. In South Asian culture, food preparation has always been considered the responsibility of the house lady-mother. Women’s recognition and personal accomplishment are pivotal in improving life functioning [1]. Health education and health promotion programs used cooking strategies to improve autonomy and sense of self among women, which boosts belief in self-worth—a symbolic medium of inclusion or exclusion [2,3]. Now being working women (sharing the economic burden caused by inflation) due to lack of time, kitchen responsibilities (especially cooking of homemade delicious foods) are being neglected. The availability of prepared food in the market serves as a substitute. Those who can afford it just place orders according to choice when needed.

Family gatherings at breakfast, lunch, and dinner times become rare. Gatherings at the dining table are not just to fulfill one’s hunger need but also serve as a learning tool, manifestation of the value system, a place of pride and connection, and a driving force of self-efficacy [3,4,5,6,7]. These gatherings provide a platform for sharing views, feelings, sorrows, worries, life histories, and socialization [8]. In other words, they are a point of strengthening social interaction and relationships, ameliorating depressive symptoms, and fostering moral transformation [3,4,9,10]. Elderly members or heads of the household guide, advise, and suggest ways to all in light of their experiences. Love and affection, honor and respect, self-control, responsibility, etc.—the pillars of healthy relationships flourish in such gatherings to fulfill physical and psychological needs and also reduce cognitive rigidity [3,4,11]. According to Cappellini and Parsons [12] (p. 7), “Sharing extraordinary meals reinforces familial bonds and perpetuates familial roles and norms”.

A revolution in the food industry and professionalism took the place of humane culture: standards and styles, preferences and choices, likes and dislikes, time, desire, direction, and communication gained importance. The technological paradigm introduces a new orientation, as a vast variety of available food has changed the patterns of consumers’ choices. Multiple factors are involved in it. Internal food factors (sensory: taste, smell, flavor, and texture and perceptual: color, nutrition and health value, portion size, and food quality) and external factors (health claims, packaging, nutritional labels of different brands) play a vital role in attracting consumers’ attention, changing perception, and preferences. Accessibility and availability of food products twist the social norms and values of dining table gatherings. These gatherings shape cultural and family identity [8,13]. Close ties of love and affection, happiness and joy, oneness, and feelings of togetherness become loose [3,14].

Social and ethical contexts have changed cultural identities consciously or unconsciously [8]. Although McDonald’s and KFC bridge other cultures, they blur out cooking traditions and practices and transform the surrounding systems (cultural context and family relations), which alters the meaning of associational embracement and creates social distancing within families [3,4,15]. In the South Asian region, food festivities remain an essential part of the culture. Specific foods are served for specific occasions and reflect cultural identity in this regard. Therefore, they maintain traditions to express associations with the culture and cultural values. Food offering in family gatherings seems to earn pride and honor, which in turn enhance happiness. Keeping in view the value and importance of traditional food, this study highlights the factors that bring changes in food choices and alter the routine of dining together, which may lead to discouraging motivation, creating disconnectedness, and affecting family ties.

### 1.1. Literature Review

The emerging food culture in the contemporary era leads towards increasing trends in cuisines. Contemporary food changes human preferences, choices, and association with cultural food identities. With the diversity and development of innovative cuisines, new dimensions of individual as well as collective identities are set out [16]. Studies indicate that the formation of identity runs from biological to cultural and from psychological to social [16,17]. The omnivore’s paradox indicates that autonomy, freedom, and adaptability oscillate between two poles of neophobia (conservatism) and neophilia (curiosity for new food) [16,17,18].

Anthony Giddens’ [19] structuration theory explains a reciprocal relationship between social actions and social structures (Figure 1). Social actions/practices are the products of social processes determined by social structures (rules and resources). The patterns of social practices represent the art of appropriate management in the flow of daily life integrated with the systems in social structures. Cultural factors, traditions, and societal ideals may influence food choices among youths. According to Giddens, these social structures (rules and resources) create the conditions for practices: constraining and enabling. Individuals’ actions and practices are determined by their choices and capacity, which reinforce changes in the social structure. Gidden’s theory supports the idea of individual action and practices. For enjoyment and fun, youths opt for integration and pleasurable experiences. In societies and families, food has always been considered a connection for relationships [20,21].

The Theory of Planned Behavior (TPB) predicts human behavior. The theory assumes that people act rationally, according to their attitude, subjective norms, and perceived behavior control [22]. In the selection of food, access and sensory needs are the predominant factors of human behavior that determine actions. The behavioral intentions of individuals refer to the likelihood of purchasing and attitudes related to food taste being better, healthier, and more available [23,24]. Youths under food-related emotions follow prevalent trends. Interactions determine intentions in the selection of foods.

The intentions behind buying behavior or the selection and purchasing of prepared food items instead of traditional homemade food depend on income, education, out-of-station jobs, marital status, knowledge (perceptions and prior experience), convenience, and time [25]. A significant factor is the increasing participation of women in the workforce. In nuclear and even in joint family systems, where male/female members are working, they do not have time for cooking or eating together. People who live in other cities because of jobs, living arrangements, workload pressures, travelling, and time constraints deprioritize cooking and eating together. Engagements have changed the preferences of consumption norms [26]. Literature indicates the combined influence of these structural components of relationships [19]. To promote healthful dietary behaviors, structural interventions were suggested by increasing the availability of social relationships and reducing social isolation [27].

In day-to-day social life, the psycho-biological mechanism of healthful dietary behavior needs much consideration. To understand eating patterns, the socio-cultural context is also important. Concerns about food safety and health consciousness shape food and feeding practices that stem from social ties/relationships. Food and feeding practices symbolize, reinforce, and reproduce social relations and divisions in the social order [28]. Studies indicate that eating practices are embedded in the context of family relationships [20,29,30]. The literature on “health and social behavior” indicates the following six key determinants/drivers that contribute to an individual’s food choices:
Biological determinants such as hunger, appetite, and taste;Economic determinants such as cost, income, availability;Physical determinants such as access, convenience, education, skills, and time;Social determinants such as class, culture, and social context–ethnicity, advertising;Psychological determinants such as mood, stress, and guilt;Attitudes, beliefs, and knowledge about food [31].

Globalization and modernization have introduced a new horizon: culture, tastes, flavors, and availability, along with consumer preferences and choices. New horizons of dietary patterns tie individuals with a larger community (a factor of social cohesion). Although diversity denies human aspects of tangibility, resemblance, durability, connection, and profundity (the cultural dimension of food), traditional or national foods have always been resistant to change (the cultural dimension of food), as there are many cultural taboos, rituals, and traditional values attached to them, along with positive health outcomes [32]. Choice of food is influenced by self-identity [33]. Food is used as a symbolic relationship, association, and sense of attachment with culture [16]. Memories of tastes from past events remind individuals of a powerful connection with food and culture. These memories sometimes connect the past with the present. Shared experiences reinforce the sense of belongingness and connect individuals with their cultural heritage.

In the political economy of food, the social context of food consumption moves towards monopoly struggle. Bourdieu [34] highlighted the distinct demonstration of food consumption with lifestyle (normative sets of practices), although cultural variations exist in societies. McDonald’s, KFC, and other prepared foods have influenced local food cultures. Media advertisements have evoked awareness among the masses, i.e., constructed a new social reality. It sets out a new fashion of food choice, preference, and symbolic identity. Quick delivery with fresh tastes, smells, and flavors attracts people, especially young ones, more towards it. In other words, the new industry creates a new culture or class—a transition from home-based food to contemporary cuisines. It is a change of identity from traditional to modern. This shift creates a point of thinking for those who love their traditional foods: how to preserve their food heritage, identity, and lineage. Margaret Mead (American anthropologist) mentioned that food connects family and friends. Gifts of food, a symbolic expression of emotional connection, bring people closer. Bowen’s family systems theory [35] indicates that family is an emotional unit where members are intensely connected emotionally [36].

The presentation of contemporary food has introduced new patterns of production and consumption [37,38]. Although palatable contemporary cuisines exhibit the contributions of chefs for a variety of cross-modal influences of taste, smell, flavor, visual features, shape, and texture [39], the symbolic presentation of food marks a transition from nature to culture and to society—a structural approach as well as a mark of power in terms of social prestige. The symbolic presentation of food instigates temptation and pleasurable sensory experiences. Studies indicate food as a factor of effective social aggregation (the cultural dimension of food) that works to shape the cultural models of society—a way of transmitting values and standards [40]. Food flavor and tastes express cultural identity. The heterogenization of food blurs out the ties of social relationships as well as social identity. Orderly interaction reflects weak social bonding between people. It evokes the question, how do you preserve and restore the identities and the central position of cultural and social dimensions?

The promotional strategy of social and print media about food products imposes reflections on the recent developments in the variety of food—offering a way of adoption, increasing demand, and maximizing the generation of wealth. The competitive price and convenience offer a way to identify modes of consumption with an increasingly fast and optimistic frenetic average lifestyle, including employment and how food is being consumed. The growing industrialization and mass production, and the establishment of supermarkets and food chains, are heading towards the connection of diet and health, nutritional aspects, food safety, process control, consumer choices, and access to food.

Worldwide, fast food has offered a new way of eating and food consumption. In the context of changing gender roles, forms of trends (less cooking and more prepared foods), and better food availability, there is a change in consumer behavior [41]. In other words, changing behavior disorients people from their ties with the importance of traditional foods having alimentary identity as well as a gradual loss of social ties (the cultural dimension of food). Therefore, in the changing world, there is a need to redefine and reinforce the dimension of food with social relationships, return to the taste and pleasure of traditional ritual traits, and recover the distortion of time and space in our lifestyle.

### 1.2. Research Questions

The research questions for this study are:
What factors influence food choices towards contemporary cuisine?How does the transformation of food choices influence traditional food?How does traditional food affect family ties?

## 2. Materials and Methods

A cross-sectional quantitative survey research method was used in this exploratory study. A systematic random sampling procedure was adopted. To obtain youth views, among public sector universities of Lahore, the University of the Punjab was selected. Among social sciences and humanities, the Institute of Social and Cultural Studies was selected. To investigate the answers to research objectives, data were collected from male and female students (*n* = 260). The questionnaire was based on the Dana-Farber Cancer Institute’s Eating Habits Questionnaire [42], Food Frequency Questionnaire (FFQ) [43], and Food Choice Questionnaire (FCQ) [44]. The questionnaire was modified according to the needs of the study. The questionnaire had four sections: demographic profile, factors shaping food choices (37 questions), kitchen culture (20 questions), and family relationships/ties (35 questions).

The questionnaire included baseline information on age, gender, education, and income, with the following sections:


**
*Preferences/choices attitude towards cuisines:*
**
Health consciousnessConsumer knowledgeBetter taste than conventional foodTrendsPrice/costSituational convenienceAvailability/social pressureCultural familiarity sensory attributes



**
*Homemade/traditional foods:*
**
Family recipesMemoriesFood flavorsElusive ingredientsHome dishes



**
*Family ties:*
**
Dining togetherAppreciationEncouragementSharing of feelings/worriesShopping for vegetables and fruitsCelebrationsMemoriesConnectedness/sense of belongingnessCreativityTraditional foodCareQuality of relationshipWell-being


A Likert scale (0—Not important/Never to 3—Very important/Often) was used in the response categories. For data analysis, descriptive (frequency and percentages) and inferential statistics (factor analysis and multivariate linear regression) were applied to the dataset to examine the relationships. Statistical significance was determined at *p* < 0.05.

## 3. Results

### 3.1. Sample Characteristics

Out of *n* = 260, 106 were males (41%), whereas 154 were females (59%); 48% were living in urban, 26% in semi-urban, and 26% in rural areas; 57% were graduates and 43% were post-graduates; and 37% of respondents’ family income was PKR 150,001 and above (Table 1).

### 3.2. Factor Analysis

A principal component analysis was carried out on responses to the 23 items. This found that there were four principal components with eigenvalues greater than one that explained 65.21% of the variance in the data. The correlation matrix revealed fairly high correlations between the variables “homemade/traditional food” and “family ties”, which indicates that the hypothesized model appears to be appropriate. The Kaiser–Meyer–Olkin measure of sampling adequacy was 0.912, and Bartlett’s test of sphericity indicated a Chi-square value of 3604.301 with a significance value <0.001, which indicates high sampling adequacy. Thus, the hypothesis that the correlation matrix is an identity matrix is rejected.

Four principal components (eigenvalue 65.21%) explained the variance in the data. The scree plot shows that a four-factors model is sufficient to represent the data (Figure 2 and Table 2).

Principal axis factoring was used to extract four factors by using the varimax method for rotation procedure. Loadings greater than 0.35 in magnitude were regarded as salient for the purposes of interpretation. Factor loadings are shown in Table 3.

All items exhibited the salient loading/determining factors. No item showed salient loading on more than one factor. From contemporary cuisines only one item, “tastes good”, exhibited salient loading. From the homemade/traditional food scale, twelve items showed salient loading: livelihood/gorgeousness, enjoyment, traditional food on special occasions, improved concentration and reduced stress, liking family recipes and food ingredients, preferring homemade food/traditional food, liking its flavors, and dining together. From family ties, ten items showed salient loading: space for conversation; strengthening relationships, closeness, and connectedness; feeling satisfied; improved quality of relationships; and enhanced well-being. Cronbach’s coefficient alpha was used to analyze the reliability of the extracted scales, which showed a satisfactory level (varied from 0.62 to 0.85).

### 3.3. Regression Analysis: Effects of Contemporary Cuisine Choices on Traditional Foods

A regression analysis was used to see the influence of contemporary cuisine choice on homemade/traditional foods. The predictor variable of homemade food explained 23% of the variance in the dependent variable of food choices and contemporary cuisine. The adjusted R-square indicates the generalization of results, which is satisfactory (Table 4).

The computed F-statistic was 78.08, with an observed significance level of less than 0.05. Thus, the hypothesis that there is no linear relationship between the predictor and dependent variable is rejected (Table 5). In other words, a relationship exists between the variables.

The Beta coefficient was shown to be positive and statistically significant at the 0.05 level. Thus, the results found an influence of food choices of contemporary cuisines on the preferences of homemade food (Table 6).

### 3.4. Regression Analysis: Effects of Contemporary Cuisine Choices on Family Ties

The predictor variable of family ties explained 31% of the variance in the dependent variable of food choices of contemporary cuisines. The adjusted R-square indicates the generalization of the results, which is satisfactory (Table 7).

The computed F-statistic was 8.29, with an observed significance level of less than 0.05, which provides a better fit, i.e., the model is significant (Table 8).

The beta coefficient was shown to be positive and statistically significant at the 0.05 level. Thus, the more food choices of contemporary cuisines there are, the weaker the family ties. The increasing trends of contemporary cuisines in food choices is weakening family ties (Table 9).

## 4. Discussion

One of the study objectives was to find out the factors influencing food choices towards contemporary cuisine. Anthony Gidden’s theory highlights how interaction and resources contribute to shaping the actions regarding the selection of foods. Food-related emotions influence ideas of tasty foods. Under the Theory of Planned Behavior (TPB), the study finding is in line with the findings of Yin et al. [24] that in “the selection of food, access and sensory needs are the predominant factors of human behavior that determine actions. The behavioral intentions of individuals refer to the likelihood of purchasing attitude related to food taste being better, healthier, and available”. For food choices, only the factor “tastes good” attracts people [18]. Among university students, it is a common social practice or fashion to get together, while the general preference is to dine outside the home. By observing the patterns of previous practices of seniors, they prefer to dine outside for the purpose of having fun, pleasure, and enjoyment to celebrate an event. It might be considered convenient, and more time is spent to have fun. This reinforces a change in daily routines, endorsed by Anthony Giddens [19] in their structuration theory: “The patterns of social practices represent the art of appropriate management in the flow of daily life integrates with the systems in social structures”. Fun and pleasure lead them to bring changes in food choices for happy moments. Happy moments enable them to ameliorate worries for a short time. Social practices with a fun motive do not impact the purchasing decision of organic or inorganic food, in contrast with the study of Padel and Foster [25]. Getting together and dining out are rare. The selection of contemporary cuisines for fun reinforces a change in the social structure [19,20,21]. A study by Gander et al. indicates that happiness ameliorates depressive symptoms and increases well-being [9].

Generally, students prefer homemade/traditional food, considering connectivity and cultural identity, as indicated by Yamamoto [16]. Food choice practices have always been part of family feeding practices that embrace nutritional value. Traditional foods play a role in building dietary choices, innate preferences, and perceptions. They also develop the sustainability of the selection of food, such as taste, digestion, and nutritional values (energy, curative, and preventive). Nutritional values focus on the food selection [20]. The homemade/traditional food, recipes, flavors, ingredients, and dinning traditions found most significant in this study are a source that create bonds. These practices encourage the sense of trust, belongingness, and closeness among family members [6,8,9,10]. While cooking and dining together, family members feel connectedness, closeness, and satisfaction [6,8,9,10,11]. These practices provide a ladder to improve not only the quality of relationships but also to enhance well-being [11,12]. Dietary knowledge contributes to and continues the social cohesion and traditions. Dietary traditions provide a sense of differentiation and identity of “what is good for health”, “harmful food”, “sequence/order to be eaten”, “vegetables”, and “prohibited food”, etc.—dietary wisdom. Homemade/traditional food engaged and hooked family members together—a representation of cultural identities and culinary traditions [16]. Preparation of food on special occasions with special ingredients and flavors along with the process embraced with traditions are considered a good sign of social integration. Practices of cooking traditional food preserves knowledge, information, and identity [45,46,47]. Eating together connects family members with one another. Eating or dining together provides a platform for conversation, discussion, and sharing of views and ideas.

This study finds the following factors about family ties:
Creates a space for conversation.Creates bonds (strengthens relationships by encouraging a sense of trust, belongingness, and closeness).Through dining and cooking, I feel connected with my family.Cooking and dining traditions are an important part of my family.Dining together is a way to bring people together.I feel satisfied when I eat at home.Dining and cooking improve the quality of relationships.It enhances my well-being.Cooking and dining affected my cultural identity development.Cooking and dining together affected my family relationships.

Deviation from homemade food choices alters the societal norms and creates distance among family members, ultimately weakening family bonds/ties. The gap between cultural and societal norms with family ties needs consideration. The health psychology research only highlights that cultural and societal norms influence food choices [48] and ignores family relationships. The results are also in line with previous research that eating practices are embedded in the context of family relationships [20,29,30]. In families, the liking of traditional foods along with cultural dimensions and the importance of family relationships exists. Other than family ties, food enhances well-being [49]. The study results indicate that traditional foods—practices and preferences of food choices/pleasures—are a source that ties family relationships.

### 4.1. Implications and Recommendations

To maintain the cultural identity and traditions of homemade food, innovative ideas are helpful to promote the knowledge of nutritional importance. This, in turn, provides a way towards healthy food choices for well-being. Homemade/traditional food is pivotal to highlight in terms of its importance in relation to connectedness, which is essential for family ties. It would be valuable to incorporate the significance of social and psychological aspects of family connectedness with food choices. The adoption of strategies to promote awareness among youth of homemade/traditional food and its connectedness with wellness may benefit social well-being. Grandmothers’ recipes, tastes, and traditional family foods are the key to healthy relationships. They develop identities among and between families. Multiple aspects should be considered for future research for a better understanding with respect to social class in different castes. Comparison of rural homemade/traditional foods with respect to contemporary cuisines choices may enhance the dynamics of preferences. Diverse samples may increase the generalizability. Finally, the study findings could motivate young ones to consider the value of homemade food to maintain health and social well-being.

### 4.2. Limitations

First, a larger sample may increase the generalizability of the findings. Secondly, a social cultural comparison may increase the variability of the findings. A diverse sample in terms of age or intergenerational perspectives and objective measures would contribute to or may enhance the opening up of new insights for a better understanding. Data from other disciplines may demonstrate causal relationships. Cultural diversity may impact the results as well.

## 5. Conclusions

The results indicate that contemporary cuisines have successfully introduced “good tastes”. People buy contemporary cuisines/food occasionally just because of taste. In daily routines, people prefer homemade food. By engaging in these practices, they regulate their kitchen values in terms of traditional foods. Cooking and dining together strengthen family relationships and enhance well-being. It encourages and motivates connectedness and closeness. Multivariate results confirmed that age and education had a significant relationship with traditional foods. Globalization and media play a role in introducing sensory appeal among youth. Those who can afford to buy such items. Homemade food (traditional/routine) fulfills all the key determinants of food choices, as indicated in “health and social behavior”, such as biological, economic, physical, social, and psychological factors, and attitudes, beliefs, and knowledge about food. The correlation results indicate that these practices not only maintain food choices but also strengthen family ties.

## Figures and Tables

**Figure 1 nutrients-16-04126-f001:**
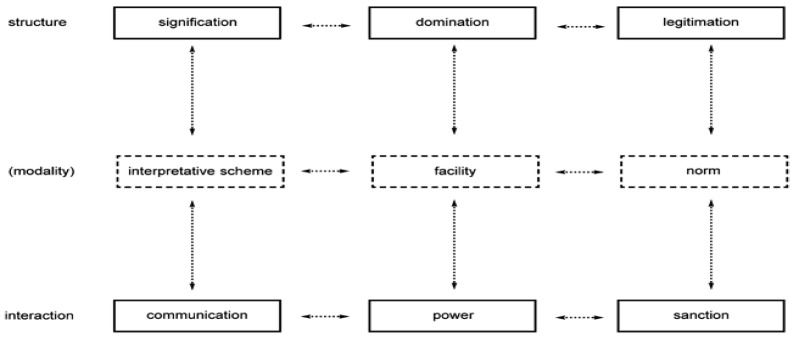
Representation of structuration theory: modalities linking social structures with social interaction (adapted from [19]).

**Figure 2 nutrients-16-04126-f002:**
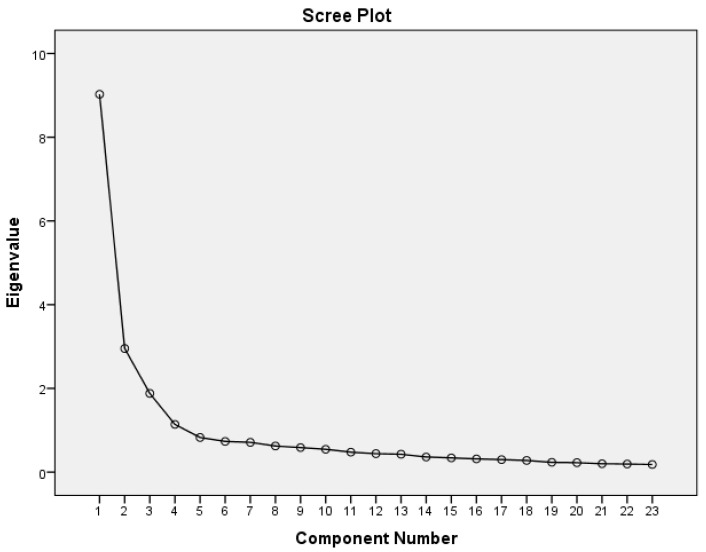
Scree plot.

**Table 1 nutrients-16-04126-t001:** Sample characteristics.

Variables		*n*	%
Gender	Male	106	40.8
	Female	154	59.2
Age (in years)	18–29	209	80.4
	30–41	27	10.4
	42–54	20	7.7
	55 and more	4	1.5
Area	Urban	125	48.1
	Semi-urban	68	26.2
	Rural	67	26.2
Education	Graduate	147	56.5
	Post-graduate	113	43.5
Family income (in PKR)	50,000–100,000	77	29.6
	100,101–150,000	86	33.1
	150,001 and more	97	37.3

**Table 2 nutrients-16-04126-t002:** Total variance explained.

	Total	% of Variance	Cumulative %	Total	% of Variance	Cumulative %	Total	% of Variance	Cumulative %
1	9.025	39.238	39.238	9.025	39.238	39.238	7.191	31.265	31.265
2	2.953	12.840	52.078	2.953	12.840	52.078	4.318	18.773	50.038
3	1.880	8.172	60.250	1.880	8.172	60.250	1.822	7.923	57.961
4	1.140	4.958	65.208	1.140	4.958	65.208	1.667	7.248	65.208
5	0.828	3.598	68.807						
6	0.733	3.185	71.992						
7	0.711	3.090	75.082						
8	0.624	2.713	77.795						
9	0.586	2.550	80.344						
10	0.546	2.372	82.717						
11	0.476	2.071	84.787						
12	0.441	1.915	86.703						
13	0.428	1.860	88.563						
14	0.361	1.572	90.134						
15	0.340	1.480	91.614						
16	0.315	1.372	92.985						
17	0.298	1.297	94.282						
18	0.279	1.211	95.493						
19	0.235	1.020	96.514						
20	0.226	0.981	97.495						
21	0.201	0.872	98.367						
22	0.194	0.842	99.209						
23	0.182	0.791	100.000						

Extraction method: principal component analysis.

**Table 3 nutrients-16-04126-t003:** Rotated component matrix.

Variables	Component			
	1	2	3	4
Cooking activities enhance home livelihood/gorgeousness (TF)	0.818			
Find enjoyment (TF)	0.815			
Ask for traditional food on special occasions (TF)	0.811			
Improves concentration and reduce stress (TF)	0.806			
Creates a space for conversation (FT)	0.798			
Creates bonds (strengthen relationships by encouraging a sense of trust, belongingness, and closeness) (FT)	0.781			
Like family recipes (TF)	0.753			
Like family food ingredients (TF)	0.750			
Prefer homemade food (TF)	0.739			
Prefer traditional foods (TF)	0.739			
Like home food flavors (TF)	0.717			
Tastes good (FC)	0.470			
Through dinning and cooking, I feel connected with my family. (FT)		0.811		
Cooking and dinning traditions are an important part of my family. (FT)		0.798		
Dining together is a way to bring people together. (FT)		0.766		
Cooking and dining traditions are an important part of my culture. (TF)		0.722		
I feel satisfied when I eat at home. (FT)		0.720		
Dinning and cooking improve the quality of relationships. (FT)		0.675		
Enhances well-being (FT)		0.622		
Dining together is not a waste of time. (TF)			0.862	
I do not feel stressed when I eat at home. (TF)			0.855	
Cooking and dining affected my cultural identity development. (FT)				0.879
Cooking and dining together affected my family relationships. (FT)				0.873

Extraction method: principal component analysis. Rotation method: varimax with Kaiser normalization. Rotation converged in 6 iterations. Note: Each item is followed by an abbreviation indicating the scale: TF—traditional food; FC—food choices; FT—family ties.

**Table 4 nutrients-16-04126-t004:** Model summary.

R	R-Square	Adjusted R-Square	Std. Error of the Estimate
0.483	0.234	0.231	19.313

Predictors: (constant), homemade food.

**Table 5 nutrients-16-04126-t005:** Results of the ANOVA.

	Sum of Squares	df	Mean Square	F	Sig.
Regression	29,126.319	1	29,126.319	78.085	<0.001
Residual	95,489.479	256	373.006		
Total	124,615.798	257			

Dependent variable: food choices of contemporary cuisines; predictors: (constant), homemade food.

**Table 6 nutrients-16-04126-t006:** Coefficients for food choices of contemporary cuisines.

	Unstandardized Coefficients		Standardized Coefficients	t	Sig.	95% Confidence Interval for B	
	B	Std. Error	Beta			Lower Bound	Upper Bound
(Constant)	38.445	3.535		10.875	<0.001	31.483	45.407
Homemade food	0.714	0.081	0.483	8.837	<0.001	0.555	0.873

Dependent variable: food choices of contemporary cuisines.

**Table 7 nutrients-16-04126-t007:** Model summary.

R	R-Square	Adjusted R-Square	Std. Error of the Estimate
0.177	0.031	0.028	15.901

Predictors: (Constant), food choices of contemporary cuisines.

**Table 8 nutrients-16-04126-t008:** Results of the ANOVA.

	Sum of Squares	df	Mean Square	F	Sig.
Regression	2367.049	1	2367.049	8.287	0.004
Residual	72,840.562	255	285.649		
Total	75,207.611	256			

Dependent variable: family ties; predictors: (constant), food choices of contemporary cuisines.

**Table 9 nutrients-16-04126-t009:** Coefficients for family ties.

	Unstandardized Coefficients		Standardized Coefficients	t	Sig.	95% Confidence Interval for B	
	B	Std. Error	Beta			Lower Bound	Upper Bound
(Constant)	62.636	3.432		18.249	<0.001	55.877	69.395
Food choices of contemporary cuisines	0.138	0.048	0.177	2.879	0.004	0.044	0.233

Dependent variable: family ties.

## Data Availability

The data presented in this study are available on request from the corresponding author.

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
