# Peer review of "Food Choices of Contemporary Cuisine and Traditional Foods: Effects on Family Ties"

_nutrients, 2024, doi:10.3390/nu16234126_

Round 1
Reviewer 1 Report
Comments and Suggestions for Authors
The introduction is unnecessarily long and detailed. It contains a lot of repeated thoughts. Although it contains many useful references, I would still consider a significant shortening.
In line 250 the brackets are unnecessary.
I agree with the necesity of Factor analysis but the tables are unnecessary. In an article a would like to see the results itself and not the working material. I understund that it was a huge work to find the explanatory factors but the results are not equeal with copying every tables from spss. I suggest not explaining the full method. Only the results: which factor explains what and how strong is the correlation. Without the full process how authors reached the results.
In 3.3 section Authors published another table from spss. I prefer highlighting the significant results and commenting all others as a non significant result. I suggest deleting all tables copied from spss and forming well-designd, comprehensible new tables or charts.
In Discussion and Conclusion I've got a really easy-to-understand version of the Results. I agree with the choosen methods and I would not recommend any further analysis only the reconsideration of publishing the results. A significant reduction in presented tables and values would improve the quality of this paper. A new structure of the results (and introduction) is essential in my opinion.
Author Response
The introduction is unnecessarily long and detailed. It contains a lot of repeated thoughts. Although it contains many useful references, I would still consider a significant shortening.
> We have added the heading “1.1 Literature review”. In addition, we also shortened the introduction section a bit.
In line 250 the brackets are unnecessary.
> We have deleted the brackets.
I agree with the necesity of Factor analysis but the tables are unnecessary. In an article a would like to see the results itself and not the working material. I understund that it was a huge work to find the explanatory factors but the results are not equeal with copying every tables from spss. I suggest not explaining the full method. Only the results: which factor explains what and how strong is the correlation. Without the full process how authors reached the results.
> We excluded the unnecessary tables. However, please note that the tables were not directly copied from SPSS. We used the format of the journal.
In 3.3 section Authors published another table from spss. I prefer highlighting the significant results and commenting all others as a non significant result. I suggest deleting all tables copied from spss and forming well-designd, comprehensible new tables or charts.
> We excluded several tables. However, we disagree with the recommendation to show only significant results. This is not good scientific practice, because also not significant results are relevant results.
In Discussion and Conclusion I've got a really easy-to-understand version of the Results. I agree with the choosen methods and I would not recommend any further analysis only the reconsideration of publishing the results. A significant reduction in presented tables and values would improve the quality of this paper. A new structure of the results (and introduction) is essential in my opinion.
> We have rewritten the discussion section. Implications and limitations have been added.
Reviewer 2 Report
Comments and Suggestions for Authors
This study investigated the factors influencing food choices in contemporary cuisine and the impact of kitchen values on family ties. However, several deficiencies can be identified from the perspective of academic review.
1.The abstract suffers from a lack of clarity and focus, making it challenging for readers to grasp the main thrust of research. The opening sentences are overly broad and contain a list of topics, such as technological advancements and social media impacts, without a clear linkage to the study's primary goal. This dilution of focus detracts from the significance of the research question. In summary, the abstract could greatly benefit from improved clarity, a more specific articulation of objectives, a more comprehensive description of methods and results, a stronger conclusion, and coherent integration of theoretical frameworks.
2. The study only identifies "taste" as a significant factor influencing contemporary cuisine choices, overlooking other crucial factors like price, convenience, cultural influences, and health concerns. Therefore, more comprehensive investigation is required.
3. The manuscript does not sufficiently emphasize its unique contribution to the field. A thorough literature review examining existing research on food choices, family ties, and cultural influences on diet is lacking. This weakens the contextual understanding.You need to go beyond describing a series of relevant references and tell us how your interpretation of the literature shows the gaps that exist, and how the proposed approach to the literature brings about novel opportunities to reinterpret the literature that will advance our understanding in the field. In addition, more updated studies may have been used.
4. The use of a modified questionnaire without clearly defined modifications raises questions regarding the instrument’s validity and reliability. The details of these modifications are therefore crucial.
5. This study focused on students from a single public-sector university in Lahore, Pakistan. This limits their generalizability to other demographic and geographic locations. The sample size was not specified for a broader context but only for male and female students.
6. The manuscript mentions structuration theory and the theory of planned behavior, but lacks a thorough discussion of how these theories guided the study design and interpretation of results.
7. The descriptions of the factor analysis, correlation, and multivariate linear regression analyses are too brief and lack sufficient detail regarding the statistical assumptions and model fit.
8. The interpretation of the statistical results must be strengthened. The manuscript should delve deeper into the practical implications of the findings and explain the unexpected or counterintuitive results.
9. The Discussion section should acknowledge and address potential confounding variables that could influence the relationship between study variables.
10.This study relied heavily on correlational analyses, which cannot establish causal relationships. The authors acknowledge the limitations of the present study. The text notes significant relationships but does not discuss the strength of the correlation coefficients.
11. Although the study mentioned testing relationships, the specific hypotheses tested were not clearly stated.
12.The reporting of p-values and statistical significance is inconsistent throughout the Results section. Standardize to present p-values in the same manner.
13. The tables and figures require more detailed captions that fully explain their contents, and the labels should be clear. The visual aids appeared to be truncated and incomplete.
14.These conclusions lack further research recommendations.
Author Response
This study investigated the factors influencing food choices in contemporary cuisine and the impact of kitchen values on family ties. However, several deficiencies can be identified from the perspective of academic review.
> Thank you very much for the feedback which we have incorporated.
1.The abstract suffers from a lack of clarity and focus, making it challenging for readers to grasp the main thrust of research. The opening sentences are overly broad and contain a list of topics, such as technological advancements and social media impacts, without a clear linkage to the study's primary goal. This dilution of focus detracts from the significance of the research question. In summary, the abstract could greatly benefit from improved clarity, a more specific articulation of objectives, a more comprehensive description of methods and results, a stronger conclusion, and coherent integration of theoretical frameworks.
> We have rephrased the abstract to focus more on clarity. We narrowed it down to the research questions and integrated results within the theoretical framework.
- The study only identifies "taste" as a significant factor influencing contemporary cuisine choices, overlooking other crucial factors like price, convenience, cultural influences, and health concerns. Therefore, more comprehensive investigation is required.
> We used factor analysis to meet the requirement of research question 1.
- The manuscript does not sufficiently emphasize its unique contribution to the field. A thorough literature review examining existing research on food choices, family ties, and cultural influences on diet is lacking. This weakens the contextual understanding. You need to go beyond describing a series of relevant references and tell us how your interpretation of the literature shows the gaps that exist, and how the proposed approach to the literature brings about novel opportunities to reinterpret the literature that will advance our understanding in the field. In addition, more updated studies may have been used.
> We covered all these gaps in the discussion section.
- The use of a modified questionnaire without clearly defined modifications raises questions regarding the instrument’s validity and reliability. The details of these modifications are therefore crucial.
> We have added Cronbach’s alpha values.
- This study focused on students from a single public-sector university in Lahore, Pakistan. This limits their generalizability to other demographic and geographic locations. The sample size was not specified for a broader context but only for male and female students.
> We have added the limitations section.
- The manuscript mentions structuration theory and the theory of planned behavior, but lacks a thorough discussion of how these theories guided the study design and interpretation of results.
> We integrated theories in the discussion.
- The descriptions of the factor analysis, correlation, and multivariate linear regression analyses are too brief and lack sufficient detail regarding the statistical assumptions and model fit.
> We have added the description.
- The interpretation of the statistical results must be strengthened. The manuscript should delve deeper into the practical implications of the findings and explain the unexpected or counterintuitive results.
> We explained the results in detail.
- The Discussion section should acknowledge and address potential confounding variables that could influence the relationship between study variables.
> In the discussion section, we have added some comments on potential factors which may confound the results.
- This study relied heavily on correlational analyses, which cannot establish causal relationships. The authors acknowledge the limitations of the present study. The text notes significant relationships but does not discuss the strength of the correlation coefficients.
> We have deleted the correlational analyses and instead provided results on the strength of coefficients.
- Although the study mentioned testing relationships, the specific hypotheses tested were not clearly stated.
> We have redone the analysis.
- The reporting of p-values and statistical significance is inconsistent throughout the Results section. Standardize to present p-values in the same manner.
> The reporting has been made consistent.
- The tables and figures require more detailed captions that fully explain their contents, and the labels should be clear. The visual aids appeared to be truncated and incomplete.
> We revisited the captions and explained the content.
14.These conclusions lack further research recommendations.
> We have added information on further research recommendations in the discussion section, because it fits better there.
Reviewer 3 Report
Comments and Suggestions for Authors
Introduction Section:
- Page 5, Lines 195-198: The research objective is clear, focusing on how contemporary food choices impact family bonds. However, the specific research questions are not well-defined. The hypotheses should be further elaborated, clearly explaining how contemporary food choices affect family values and relationships.
- Page 4, Lines 35-49: The background discusses the impact of economic development and social media on eating behaviour, but it is too general. It lacks specific discussion on changes in family roles and eating habits in South Asian cultural contexts. A more in-depth analysis of the cultural background in this region is recommended.
- Page 4, Lines 49-53: The significance and motivation section highlights the effects of social change on family and eating behaviour. However, it should place greater emphasis on how the study fills existing gaps in academic research and further explain its practical applications for policy-making or health promotion.
Literature Review Section:
- Page 5, Lines 171-194: The literature review focuses on family eating behaviour and cultural identity but lacks sufficient discussion on the conflicts between contemporary and traditional food choices, as well as the impact of cultural and social structure changes. More comparative studies from different cultural backgrounds should be included.
- Page 4, Lines 99-114: The literature review does not clearly identify the main research gaps in the existing literature. It is recommended to emphasize how contemporary food choices affect family values and relationships, and to point out the lack of discussion on this topic in previous studies.
- Page 5, Lines 195-197: The article references structuration theory and the theory of planned behaviour, but the application and explanation of these theories are not thorough enough. The link between contemporary food choices and family eating habits is not clearly explained. The theoretical support should be strengthened, explaining how these theories guide the research.
Methodology Section:
- Page 5, Lines 199-202: The study employs a cross-sectional survey design, which is appropriate for exploratory research but cannot capture long-term changes. This limitation should be explicitly discussed in the article.
- Page 5, Lines 202-205: The sample selection is limited to public university students in Lahore, which may not represent a broader societal group. The sample range should be expanded to reflect perspectives from different social strata.
- The operational definitions of variables need further clarification, particularly regarding “family values” and “family bonds.” Providing clearer operational definitions would improve the replicability of the research.
- Using SPSS for data analysis is appropriate, but the connections between variables should be explained in greater detail, and it should be clarified how the multiple regression analysis supports the research hypotheses.
Results Section:
- Page 8, Lines 248-251: The data presentation is generally clear, but the interpretation of some results needs to be more in-depth, particularly regarding the interrelationship between variables. This would help readers better understand the significance of the findings.
- Page 8, Lines 247-264: The statistical analysis methods are correct and convincing, but more emphasis should be placed on the relevance and significance of the relationships between key variables, making the research findings clearer.
Discussion Section:
- Page 16, Lines 328-334: The discussion section engages in limited dialogue with the literature. The research findings should be compared more extensively with existing literature to explore whether the results support or challenge established theories, particularly regarding how contemporary food choices affect family relationships.
- Page 16, Lines 338-341: The current discussion on the practical applications and theoretical contributions of the research is not deep enough. The study's contributions to health promotion and cultural preservation should be further elaborated.
Conclusion Section:
- Page 17, Lines 390-395: The conclusion reasonably summarizes the research findings, but the core findings, especially how contemporary food choices influence family values and relationships, should be more succinctly outlined.
- Page 17, Lines 396-399: The practical significance of the research should be emphasized more, such as how the study contributes to health promotion, cultural preservation, and the improvement of family life quality.
Limitations and Future Research Section:
- Page 16, Lines 344-350: The discussion of research limitations is not comprehensive. The article should address the issues of sample representativeness, the limitations of the cross-sectional design, and the potential influence of cultural background differences, helping readers better understand the generalizability of the findings.
- Page 17, Lines 399-402: The suggestions for future research are somewhat vague. Specific directions should be proposed, such as comparative studies across different cultural backgrounds, genders, or age groups, or a more in-depth exploration of the long-term effects of contemporary food choices on the evolution of family roles.
Author Response
Introduction Section:
Page 5, Lines 195-198: The research objective is clear, focusing on how contemporary food choices impact family bonds. However, the specific research questions are not well-defined. The hypotheses should be further elaborated, clearly explaining how contemporary food choices affect family values and relationships.
> We have reframed the research questions for clarity. Efforts have been made to elaborate on the relationships between the variables.
Page 4, Lines 35-49: The background discusses the impact of economic development and social media on eating behaviour, but it is too general. It lacks specific discussion on changes in family roles and eating habits in South Asian cultural contexts. A more in-depth analysis of the cultural background in this region is recommended.
> We have added a paragraph on the specific situation in South Asia.
Page 4, Lines 49-53: The significance and motivation section highlights the effects of social change on family and eating behaviour. However, it should place greater emphasis on how the study fills existing gaps in academic research and further explain its practical applications for policy-making or health promotion.
> We have elaborated on how the study fills existing gaps in the discussion section.
Literature Review Section:
Page 5, Lines 171-194: The literature review focuses on family eating behaviour and cultural identity but lacks sufficient discussion on the conflicts between contemporary and traditional food choices, as well as the impact of cultural and social structure changes. More comparative studies from different cultural backgrounds should be included.
> We have elaborated on these issues in the fourth paragraph of the literature revies and in the second paragraph of the discussion section.
Page 4, Lines 99-114: The literature review does not clearly identify the main research gaps in the existing literature. It is recommended to emphasize how contemporary food choices affect family values and relationships, and to point out the lack of discussion on this topic in previous studies.
> We have rephrased the section to point out the research gaps.
Page 5, Lines 195-197: The article references structuration theory and the theory of planned behaviour, but the application and explanation of these theories are not thorough enough. The link between contemporary food choices and family eating habits is not clearly explained. The theoretical support should be strengthened, explaining how these theories guide the research.
> We have developed a link to both theories in the discussion section.
Methodology Section:
Page 5, Lines 199-202: The study employs a cross-sectional survey design, which is appropriate for exploratory research but cannot capture long-term changes. This limitation should be explicitly discussed in the article.
> We have referred to this issue in the limitations section.
Page 5, Lines 202-205: The sample selection is limited to public university students in Lahore, which may not represent a broader societal group. The sample range should be expanded to reflect perspectives from different social strata.
> We have mentioned this aspect in the recommendations.
Using SPSS for data analysis is appropriate, but the connections between variables should be explained in greater detail, and it should be clarified how the multiple regression analysis supports the research hypotheses.
> We have rewritten the section on data analysis for clarification.
Page 8, Lines 248-251: The data presentation is generally clear, but the interpretation of some results needs to be more in-depth, particularly regarding the interrelationship between variables. This would help readers better understand the significance of the findings.
> We have modified the analysis.
Page 8, Lines 247-264: The statistical analysis methods are correct and convincing, but more emphasis should be placed on the relevance and significance of the relationships between key variables, making the research findings clearer.
> We have added this.
Discussion Section:
Page 16, Lines 328-334: The discussion section engages in limited dialogue with the literature. The research findings should be compared more extensively with existing literature to explore whether the results support or challenge established theories, particularly regarding how contemporary food choices affect family relationships.
> We have added this.
Page 16, Lines 338-341: The current discussion on the practical applications and theoretical contributions of the research is not deep enough. The study's contributions to health promotion and cultural preservation should be further elaborated
> We have elaborated on this in the discussion section.
Page 17, Lines 390-395: The conclusion reasonably summarizes the research findings, but the core findings, especially how contemporary food choices influence family values and relationships, should be more succinctly outlined.
> We have added this.
Page 17, Lines 396-399: The practical significance of the research should be emphasized more, such as how the study contributes to health promotion, cultural preservation, and the improvement of family life quality.
> We have added this.
Limitations and Future Research Section:
Page 16, Lines 344-350: The discussion of research limitations is not comprehensive. The article should address the issues of sample representativeness, the limitations of the cross-sectional design, and the potential influence of cultural background differences, helping readers better understand the generalizability of the findings.
> We have elaborated on this in the limitations section.
Page 17, Lines 399-402: The suggestions for future research are somewhat vague. Specific directions should be proposed, such as comparative studies across different cultural backgrounds, genders, or age groups, or a more in-depth exploration of the long-term effects of contemporary food choices on the evolution of family roles.
> We have added this.
Round 2
Reviewer 1 Report
Comments and Suggestions for Authors
With the lower number of tables the presented information bacame much better compared to the first version. Some of tables are still containing every information which is not relevant and does not help the understanding of this study but they are not disturbing so I accept the decision of the authors. Of course the non-significant results are as important as the significant ones but the way of interpretation was too detailed in some cases. But I also accept the authors decision. Somehow I still feel at some points in the results this article more likely to be a statistical description than a report of the results.
Author Response
With the lower number of tables the presented information bacame much better compared to the first version. Some of tables are still containing every information which is not relevant and does not help the understanding of this study but they are not disturbing so I accept the decision of the authors. Of course the non-significant results are as important as the significant ones but the way of interpretation was too detailed in some cases. But I also accept the authors decision. Somehow I still feel at some points in the results this article more likely to be a statistical description than a report of the results.
> Thanks. We have added description to enhance readers’ understanding.
Reviewer 2 Report
Comments and Suggestions for Authors
Dear authors,
Thank you for your comprehensive response to my review comments and for addressing the suggestions provided. I have carefully reviewed the revisions and am pleased to see that my concerns have been satisfactorily addressed.
Author Response
Dear authors,
Thank you for your comprehensive response to my review comments and for addressing the suggestions provided. I have carefully reviewed the revisions and am pleased to see that my concerns have been satisfactorily addressed.
> Thank you very much for your positive feedback. Your comments helped us to improve the quality of the manuscript.
Reviewer 3 Report
Comments and Suggestions for Authors
Upon reviewing the manuscript, it appears that the sections on literature review, conclusion and discussion, as well as research limitations, do not yet meet the expected scholarly standards.
First, the literature review lacks depth in its analysis and falls short of providing a critical examination of key theories and developments in the field. This absence of comprehensive engagement with relevant literature diminishes the contextual grounding of the study and obscures its contribution to the field. It is recommended that the author expands this section by incorporating a systematic review of pertinent studies to better situate the research within the existing body of knowledge and clarify its distinct contributions.
In addition, the conclusion and discussion sections remain overly general and do not sufficiently address the research questions or elucidate the implications of the findings. A reorganization of these sections is advised, with a focus on deepening the discussion of the data analysis outcomes and directly addressing the research questions. This approach would help highlight the theoretical and practical value of the findings.
Finally, the section on research limitations lacks adequate critical reflection. It would be beneficial for the author to discuss in greater detail the limitations regarding methodology, sample data, and other constraints, along with their potential impact on the validity and generalizability of the results. Such reflection is essential for providing readers with a balanced understanding of the study’s scope and guiding future research directions.
I encourage the author to make these revisions to enhance the manuscript’s academic rigor and strengthen its overall impact.
Author Response
Upon reviewing the manuscript, it appears that the sections on literature review, conclusion and discussion, as well as research limitations, do not yet meet the expected scholarly standards.
> Thank you for your recommendations which we have addressed in the revised version.
First, the literature review lacks depth in its analysis and falls short of providing a critical examination of key theories and developments in the field. This absence of comprehensive engagement with relevant literature diminishes the contextual grounding of the study and obscures its contribution to the field. It is recommended that the author expands this section by incorporating a systematic review of pertinent studies to better situate the research within the existing body of knowledge and clarify its distinct contributions.
In addition, the conclusion and discussion sections remain overly general and do not sufficiently address the research questions or elucidate the implications of the findings. A reorganization of these sections is advised, with a focus on deepening the discussion of the data analysis outcomes and directly addressing the research questions. This approach would help highlight the theoretical and practical value of the findings.
> We have added links to theories, in this case Giddens Structuration Theory and Theory pf Planned Behavior. Furthermore, we elaborated more on the psycho-biological mechanism for healthful dietary behaviors. For the understanding of eating patterns, the socio-cultural context has been emphasized. Concerns about food safety and health consciousness shape food practices that stem from social ties/relationships.
The literature on “health and social behavior” indicates six key determinants that can contribute to an individual’s food choice. Cultural dimensions of food in relation with health, food choices and self-identity are also discussed for clarifications.
Finally, the section on research limitations lacks adequate critical reflection. It would be beneficial for the author to discuss in greater detail the limitations regarding methodology, sample data, and other constraints, along with their potential impact on the validity and generalizability of the results. Such reflection is essential for providing readers with a balanced understanding of the study’s scope and guiding future research directions.
> We have further elaborated on the limitations (e.g., sample size and social cultural aspects).
I encourage the author to make these revisions to enhance the manuscript’s academic rigor and strengthen its overall impact.
> Thank you very much for your helpful comments.
Round 3
Reviewer 3 Report
Comments and Suggestions for Authors
Thanks to the author for the revisions.